# COMPOFA: COMPOUND ONCE-FOR-ALL NETWORKS FOR FASTER MULTI-PLATFORM DEPLOYMENT

**Manas Sahni, Shreya Varshini, Alind Khare, Alexey Tumanov**
Georgia Institute of Technology
`{sahnimanas, shreyavarshini, alindkhare, atumanov}@gatech.edu`

## ABSTRACT

The emergence of CNNs in mainstream deployment has necessitated methods to design and train efficient architectures tailored to maximize the accuracy under diverse hardware & latency constraints. To scale these resource-intensive tasks with an increasing number of deployment targets, Once-For-All (OFA) proposed an approach to jointly train several models at once with a constant training cost. However, this cost remains as high as 40-50 GPU days and also suffers from a combinatorial explosion of sub-optimal model configurations. We seek to reduce this search space – and hence the training budget – by constraining search to models close to the accuracy-latency Pareto frontier. We incorporate insights of compound relationships between model dimensions to build *CompOFA*, a design space smaller by several orders of magnitude. Through experiments on ImageNet, we demonstrate that even with simple heuristics we can achieve a 2x reduction in training time[1] and 216x speedup in model search/extraction time compared to the state of the art, *without* loss of Pareto optimality! We also show that this smaller design space is dense enough to support equally accurate models for a similar diversity of hardware and latency targets, while also reducing the complexity of the training and subsequent extraction algorithms.[2]

## 1 INTRODUCTION

CNNs are emerging in mainstream deployment across diverse hardware platforms, latency requirements, and/or workload characteristics. The available processing power, memory, and latency requirements may vary vastly across deployment platforms – say, from server-grade GPUs to low-power embedded devices, cycles of high or low workload, etc.

Since model accuracies tend to increase with computational budget, it becomes vital to build models tailored to each deployment scenario, maximizing accuracy constrained by the desired model inference latency. These efficient models lie close to the Pareto-frontier of the accuracy-latency trade-off. Building such models (either manually or by searching) and then training them are resource-intensive tasks – requiring massive computational resources, expertise in both ML and underlying systems, time, dollar cost, and $CO_2$ emissions every time they are performed. Repeating such intensive processes for *each* deployment target is prohibitively expensive w.r.t. multiple metrics of cost and this does not scale.

Once-For-All (OFA) (Cai et al., 2020) proposed to address this challenge by decoupling the search and training phases through a novel progressive shrinking algorithm. OFA builds a family of $10^{19}$ models of varying depth, width, kernel size, and image resolution. These models are jointly trained in a single-shot via sharing of their intersecting weights. Once trained, search techniques can extract specialized sub-networks that meet specific deployment targets – a task that can then be independently repeated on the same trained family.

This massive search space leads to a training cost that remains prohibitively expensive. Though the cost can be amortized over a number of deployment targets, it' still significant – reaching 1200 GPU hours for OFA. The search space arises from training every possible model combination, and many

---

[1]and, therefore, dollar cost and $CO_2$ emissions
[2]Our source code is available at `https://github.com/gatech-sysml/CompOFA`

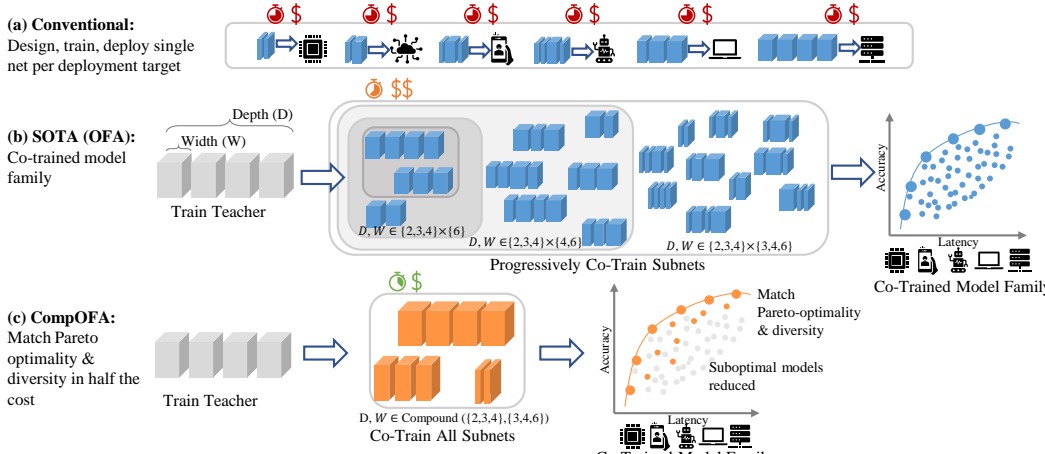

Figure 1: **(a):** Conventional methods require expensive designing & training per deployment plat-form, which is infeasible to scale. **(b):** OFA co-trains a family of subnetworks of a teacher supernet. However, combinatorial explosion of depth (D) and width (W) compels progressive, phased training requiring 1200 GPU hours. **(c):** CompOFA exploits the insight of compound couplings between D & W to vastly simplify the search space while maintaining Pareto optimality. The smaller space can be trained in half the time without phases, and gives equally performant and diverse model families

of these models may lie well below the accuracy-latency Pareto frontier, as represented in Figure 1(b). This exhaustive approach misses opportunities for any accuracy- or latency-guided exploration in such a vast space, thus suffering a clear inefficiency. These sub-optimal models not only go un-utilized but also add training interference, which necessitates a longer phased training to stabilize their simultaneous optimization. Finally, searching & extracting an optimal model from this space can only be done via indirect estimators of their accuracy and latency, as all model combinations cannot be enumerated.

On the other hand, we argue that such a large search space in unnecessary for two reasons. First, common practices as well as empirical studies (Tan & Le, 2019; Radosavovic et al., 2020) have shown that model dimensions such as depth, width, and resolution are not orthogonal – models that follow some compound couplings between these dimensions produce a better accuracy-latency trade-off than those with unconstrained settings. Informally, increasing model capacity along one dimension (say, depth) is helped by an accompanying increase along another dimension (say, width). Secondly, a much coarser latency granularity (order of 1 ms) is *sufficient* for practical systems de-ployment.

In this work we propose *CompOFA* – a model design space leveraging compound couplings between model dimensions, and demonstrate the following:

1. Utilizing the insight of compound coupling, we show that simple, easy to implement heuris-tics can capture models close to the Pareto frontier (depicted in Figure1(c). This enables us to reduce OFA's $10^{19}$ models to just 243 in CompOFA, and still train a model family with an equally good accuracy-latency tradeoff.
2. We show that this tractable design space directly reduces interference in training, which allows us to reduce training duration and cost by **2x**.
3. Once trained, CompOFA's simplicity avails itself to easier extraction that's faster by **216x**.
4. Despite the size reduction, we show that the latency granularity is *sufficient* to cover the same range and diversity of hardware targets as OFA.
5. Finally, the generality of CompOFA's insights is validated by training it on another base architecture, achieving similar gains.

## 2 RELATED WORK

Efficient neural network design has been an active area of research due to the high computational complexity of CNNs. NAS is increasingly used to guide or replace previously manual design pro-

cesses. Early NAS techniques (Zoph et al., 2018; Zoph & Le, 2016) sampled several architectures and trained them from scratch every time, making them extremely compute-hungry. The technique of weight-sharing has emerged as one way to address these inefficiencies (Berman et al., 2020; Bender et al., 2018; Brock et al., 2017; Guo et al., 2019; Liu et al., 2018; Pham et al., 2018). These methods slice candidate sub-networks from a larger *super-network*, thereby sharing weights between them. Simultaneously, latency-guided NAS methods (Tan et al., 2019; Cai et al., 2018; Berman et al., 2020; Stamoulis et al., 2019; Wu et al., 2019) have sought to incorporate model complexity into their search to find efficient models for a given target latency on a given hardware target.

Nevertheless, these methods yield a single model per run – both search *and* training must be repeated for new deployment targets. With compute costs reaching as high as $O(10^4)$ GPU hours, this linear scaling is infeasible for an ever-growing need for multi-platform, multi-latency deployment. Once-For-All (OFA) (Cai et al., 2020) proposed to reduce this cost by using weight-sharing for a large model *family* that collectively supports diverse range of latencies. They perform one-time training of $O(10^{19})$ sub-networks, which can then be independently searched to support a given deployment target later, thus amortizing the training cost. However, this cost is still prohibitively expensive, reaching 1200 GPU hours. The unnecessarily large search space complicates both training and searching, stemming from an uninhibited combinatorial explosion of model configurations.

On the other hand, empirical studies on neural network design spaces (Tan & Le, 2019; Radosavovic et al., 2020) have recently shown that model dimensions (e.g., depth, width, resolution) are not independent – underlying relations between them can be used to obtain an optimal accuracy-latency trade-off. In other words, the number of degrees of freedom in the architecture search space can be reduced *without* loss of Pareto optimality. This insight forms the basis of our work to constrain the design space of OFA networks while maintaining the same quality (w.r.t. accuracy) as well as diversity of models with reduced train and search costs.

Dynamic Neural Networks with weight-sharing across models of varying latencies have been explored before (Yu et al., 2018; Yu & Huang, 2019) but with much fewer models (e.g. 4 in Yu et al. (2018)) and few dimensions (e.g. only width) which make for much sparser and narrower support for diverse latency targets. We explore a middle ground between these works and Cai et al. (2020) to build design spaces that are tractable yet sufficiently diverse to support many latency targets and varying in multiple model dimensions.

Yu et al. (2020) proposed replacing OFA's multi-stage training by a single-stage one, challenging the usual practive of progressive training in one-shot NAS. However, its training cost remains high at over 2300 TPU hours for $O(10^{12})$ models. Contemporary to our work, Wang et al. (2020) point out a similar wasted computation on sub-optimal models in one-shot NAS, and use attention mechanisms to push the Pareto front. Our approach instead focuses on achieving the *same* Pareto frontier as OFA with a smaller budget. We also emphasize architectural insights to show that an intractably large cardinality is not necessary in the first place – simple heuristics identify good models while enabling a host of other cost savings.

## 3 MOTIVATION

### 3.1 DESIGN SPACE PARAMETRIZATION

Consider a network architecture $\mathcal{N}$ composed of $m$ micro-architectural blocks, $B_1, B_2, \ldots, B_m$. Each block is parametrized by its depth, per-layer width, and per-layer kernel size as $B_i(d_i, W_i, K_i)$. Here $d_i$ denotes the number of layers (depth) in the block, and $W_i$ and $K_i$ are lists denoting the width & kernel sizes of each of these $d_i$ layers.

Once-For-All(Cai et al., 2020) builds a family of networks $\mathcal{N}_1, \mathcal{N}_2, \ldots$ of varying accuracies and latencies. The weights of common layers and channels of a block $B_i$ are shared across all the networks. The "block" used in OFA is the Inverted Residual from MobileNetV3 (Howard et al., 2019) and hence "width" here refers to the channel expansion ratio. $d_i, w_{ij}, k_{ij}$ are sampled independently from sets $D = [2, 3, 4], W = [3, 4, 6], K = [3, 5, 7]$ respectively.

In OFA, each of these dimensions is treated as orthogonal. Hence, the resulting number of possible networks is enormous, with $O(10^{19})$ models for $m = 5$ blocks(Cai et al., 2020).

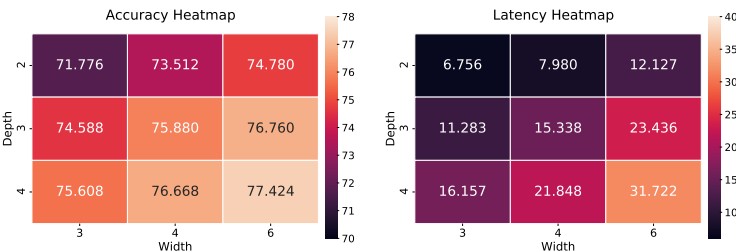

Figure 2: Accuracy and latency heatmaps for varying uniform depth and expansion ratios and fixed kernel size=5, as measured in MobileNetV3 architecture. Latency is measured in ms on NVIDIA RTX 2080 Ti GPU with BS=64.

## 3.2 COMPOUND RELATION OF MODEL DIMENSIONS

The combinatorial explosion from just 3 independent model dimensions of $D, W, K$ yields a model family with an enormous number of models. While the aim for jointly training "every possible model" in this large design space is to support a diverse range of hardware platforms, we note the following concerns that arise from this:

1. **Model dimensions are not orthogonal**
   Scaling model dimensions like depth, width, and resolution to higher FLOP regimes is a common practice and Tan & Le (2019) systematically showed that compound relations exist between these dimensions for achieving an optimal accuracy-latency trade-off. In particular, increasing model dimensions in accordance with a compound scaling rule gives better results than doing so independently. Radosavovic et al. (2020) elevated this concept to the model *population* level and found that a *quantized linear relation* between model depth and width yielded design spaces with a better concentration of well-performing models. These works solidified the common practice of jointly increasing model depth and width. Yet, all these other sub-optimal models are still included in the search space when all dimensions are sampled independently.

2. **Large model families complicate training & searching**
   The interference between OFA's sub-networks complicates their simultaneous optimization and necessitates techniques like progressive shrinking (Cai et al., 2020), increasing training time. Further, extracting models for a desired target cannot rely on simple enumeration and instead needs to rely on predicting model accuracy/latency. This requires additional training for these accuracy predictors specific to the trained model family and collecting latency look-up tables (Cai et al., 2018). With a more tractable cardinality, we could rely on more standard, faster approaches during training and achieve "off-the-shelf" usability during search.

3. **Hardware latency differences below a certain granularity are noisy**
   Finally, the difference between unique architectures' accuracy or latency needs to be distinguishable during search. The original OFA design space covers a FLOP range of 120-560 MFLOPs and within this range there exist $O(10^{19})$ architecture choices. Even if these models were uniformly distributed into buckets of 1 FLOP, we would still have $O(10^{11})$ models in *each* FLOP target. Note that these results do not account for variable resolution, which would otherwise further multiply the number of choices at each FLOP target. On any hardware, inference time below a certain threshold is expected to be indistinguishable from noise. This threshold can vary depending on application context or hardware (e.g. it can be upto 1-5 ms in ML serving scenarios). Irrespective of the threshold, having these many models with such fine-grained differences in their compute requirement is well below the minimum error resolution at which they can be meaningfully compared. This motivates us to consider that a design space that is sparser by several orders of magnitude might still support the same density and range of deployment targets.

Figure 2 shows heatmaps comparing accuracy and latency of OFA sub-networks with uniform depth and width for a fixed kernel size. A closer look into these heatmaps, depicts a monotonically increasing gradient along the diagonal of each of these two dimensional heatmaps. This observation

showcases that when the configurations are increased along both the depth and width dimensions we retrieve a class of sub-networks that achieve a better accuracy-latency trade-off as opposed to those formulated through a single dimensional configuration change. Thereby, emphasizing the existence of a coefficient that dictates the relationship between depth and width configurations of a model. This suggests that by quantitatively conforming to this coupling between the model dimensions, we can aim toward reducing the network design space of an architecture without sacrificing its accuracy. Subsequently, this further implies that OFA is unnecessarily training a lot more models than required and we additionally show that OFA's model distribution suffers from high variance as well (Appendix A.1). This leads us toward a solution that leverages these insights to prune out redundant and ineffective sub-networks otherwise generated by the unconstrained OFA architecture.

## 4 COMPOUND OFA

### 4.1 COUPLING HEURISTIC

Motivated by the above observations, we constrain the network design space using a simple heuristic: depth and width dimensions should increase or decrease together, not independently. Specifically, in each block, whenever we sample the $i^{\text{th}}$ largest depth $d_i \in D$, we correspondingly sample the $i^{\text{th}}$ largest width $w_i \in W$ for all $d_i$ layers in the block. For instance, with $D = [2, 3, 4]$ and $W = [3, 4, 6]$, each block can have either two layers of channel expansion ratio 3, or three layers of ratio 4, or four layers of ratio 6. Once this modified search space is defined by the heuristic, the method of extracting a given model configuration remains the same – we slice out a sub-network of the specified dimensions from the largest network, thus sharing common weights between all sub-networks.

This significantly reduces the degrees of freedom in the design space. An additional consequence of this is that all layers within a block now have the same width, further reducing the architecture-count. Nevertheless, we will show in Section 5 that the design space is still diverse and dense enough to provide models for different deployment requirements.

We fix the kernel size by default, but for fair comparison we also show a variant with Elastic-Kernel. With a fixed-kernel, we create a simplified search space with kernel sizes fixed to either 3 or 5 in each block, to match the original setting used in MobileNetV3. Thereby in this design space, depth (or width) alone can fully specify a block's configuration. For 5 blocks, this yields a family of $3^5 = 243$ models. In the elastic-kernel design space, we expand the kernel size as done in OFA, sampled from $K = [3, 5, 7]$ per layer. This results in a family of $(3^2 + 3^3 + 3^4)^5 \approx 10^{10}$ models. Unless otherwise specified, we use "CompOFA" to refer to the fixed-kernel design space.

As we show in the following section, the source of our training speedup is the reduction in cardinality of the search space. The input resolution to the model affects the model inference time but does not affect the number of unique trainable architectures. Hence, we keep the resolution elastic, which allows using one architecture to support multiple latencies for "free", without increasing our search space cardinality (and soon, training budget).

### 4.2 TRAINING SPEEDUP

Cai et al. (2020) proposed a "progressive shrinking" approach to train the OFA sub-networks in a phased manner, starting with the largest model and then progressively including smaller sub-networks in training. At each batch, a different sub-network is sampled for forward/backward pass. If the expected time to complete one epoch in phase $p$ is $\mathbb{E}(t_p)$ and the training is run for $e_p$ number of epochs, then the total time to train a model family becomes $T_{family} \propto \sum_{p \in Phases} e_p \times \mathbb{E}(t_p)$.

Our goal is to reduce $T_{family}$ while achieving the same level of accuracy-latency trade-off and supporting the same range of deployment scenarios. Since we wish to train models of similar latency targets, we try to keep $\mathbb{E}(t_p)$ unchanged and target a reduction in the number of phases to reduce $T_{family}$.

While the number of models does not explicitly factor in $T_{family}$, note that it implicitly affects the count & duration of training phases – progressive shrinking was required in the first place due to interference between a large number of models. The maximum number of simultaneously trainable

Table 1: Training schedule, duration, and GPU hour comparisons for OFA and CompOFA. CompOFA reduces the training time of OFA by 50% with a fixed kernel space. The columns $K, D, W$ represent the sets of possible model dimensions (as in Section 3.1). While CompOFA allows all sub-networks to be trained after the teacher network, OFA progresses to the full search space with multiple phases of similar duration. See Appendix A.3 for CompOFA with Elastic Kernel

(a) Once-For-All (Cai et al., 2020) (Elastic Kernel)

| Phase | $K$ | $D$ | $W$ | $N_{sample}$ | Epochs | Wall Time | GPU Hours |
|---|---|---|---|---|---|---|---|
| Teacher | 7 | 4 | 6 | 1 | 180 | 28h 45m | 172h 30m |
| Elastic Kernel | 3, 5, 7 | 4 | 6 | 1 | 125 | 26h 51m | 161h 06m |
| Elastic Depth-1 | 3, 5, 7 | 3, 4 | 6 | 2 | 25 | 7h 46m | 46h 36m |
| Elastic Depth-2 | 3, 5, 7 | 2, 3, 4 | 6 | 2 | 125 | 38h 32m | 231h 12m |
| Elastic Width-1 | 3, 5, 7 | 2, 3, 4 | 4, 6 | 4 | 25 | 10h 06m | 60h 36m |
| Elastic Width-2 | 3, 5, 7 | 2, 3, 4 | 3, 4, 6 | 4 | 125 | 51h 03m | 306h 18m |
| **Total** | | | | | **605** | **163h 03m** | **978h 18m** |

(b) CompOFA (Fixed Kernel)

| Phase | $K$ | $D$ | $W$ | $N_{sample}$ | Epochs | Wall Time | GPU Hours |
|---|---|---|---|---|---|---|---|
| Teacher | 7 | 4 | 6 | 1 | 180 | 28h 45m | 172h 30m |
| Compound | − | 2, 3, 4 | 3, 4, 6 | 4 | 25 | 8h 43m | 52h 18m |
| Compound | − | 2, 3, 4 | 3, 4, 6 | 4 | 125 | 44h 47m | 268h 42m |
| **Total** | | | | | **330** | **82h 15m** | **493h 30m** |

models in CompOFA's search space is smaller by 17 orders of magnitude. With this significantly smaller family size, we find that the interference between these models is reduced to the extent that we are now able to remove progressive shrinking altogether, and instead achieve the same accuracy with all trainable sub-networks included in training (see Section 5).

In Section 5 we show that this allows us to reduce the training time by 50%. Additionally, we show an ablation to confirm that this reduction in phases is only possible due to the reduced number of models in CompOFA, and not in the original large design space of OFA.

## 5 EXPERIMENTS

### 5.1 TRAINING SETUP

We train the full-sized CompOFA network ($D = [4], W = [6]$) on ImageNet (Deng et al., 2009) using the same base architecture of MobileNetV3 as OFA. We use a batch size of 1536 and a learning rate of 1.95 to train on 6 NVIDIA V100 GPUs. All other training hyperparameters are kept the same as OFA for accurate comparison.

Next, for CompOFA with fixed kernel sizes, we unlock all permissible model configurations ($D, W = [(2, 3), (3, 4), (4, 6)]$) in one stage, as opposed to progressively adding more configurations. We train this for 25 epochs with an initial learning rate of 0.06, followed by 120 epochs with an initial learning rate of 0.18 to obtain our final network. We sample $N_{sample} = 4$ models in each batch and aggregate their gradients before each optimizer step. The full network serves as a teacher model for knowledge distillation (Hinton et al., 2015). The comparison of the possible configurations and epoch durations for each training phase is summarized in Table 1.

### 5.2 TRAINING TIME AND COST

By using just one stage of training after the teacher model (in the default fixed kernel setting), instead of multiple stages of the same duration, we are able to reduce the training cost of CompOFA by 50%. Our reproduction of OFA's training scheme on 6 GPUs takes 978 GPU hours while CompOFA-Fixed Kernel finishes in 493 GPU hours, as shown in Table 1. Table 2 shows the translation of this time reduction in terms of dollar cost and $CO_2$ emissions. Note that the total duration of an epoch with the same design space configuration is comparable, as we do not target a speedup by training smaller

Table 2: Comparing OFA and CompOFA on training monetary cost, $CO_2$ emission and average search duration. Monetary cost is based on hourly price of 1 NVIDIA V100 on Google Cloud. $CO_2$ emission estimation is based on Strubell et al. (2019). Search time is reported for an average over latency targets, without the use of latency estimators

| Method | Cost | $CO_2$ emission | Avg. Search Time |
|---|---|---|---|
| OFA | $2.4k | 277 lbs | 4.5 hours |
| CompOFA (Elastic Kernel) | $1.7k | 196 lbs | 2.25 hours |
| CompOFA (Fixed Kernel) | $1.2k | 138 lbs | **75 seconds** |

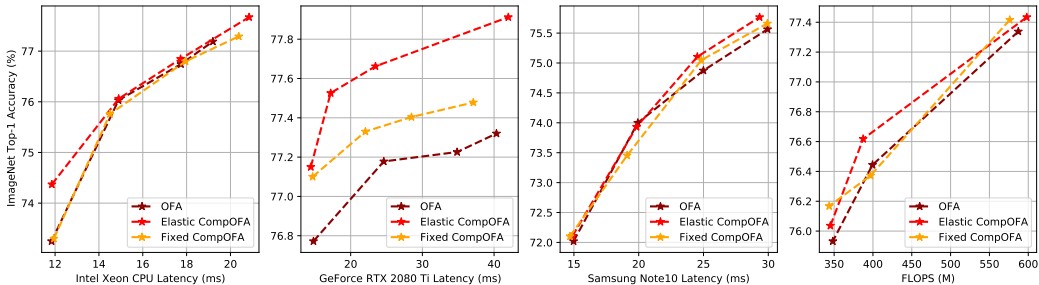

Figure 3: CompOFA networks consistently achieve comparable and higher ImageNet accuracies for similar latency and FLOP constraints on CPU, GPU and mobile platforms within 1-5ms granularities.

models. Instead, the speedup stems largely from halving the number of epochs, by way of reducing the phases for training. This reduction is in turn possible due to a smaller, constrained design space.

## 5.3 ACCURACY-LATENCY TRADE-OFF

After training CompOFA networks, we carried out an evolutionary search (Real et al., 2019) to retrieve specialized sub-networks for diverse hardware deployment scenarios, as done in OFA. The evolutionary search fetches trained sub-networks that maximize accuracy subject to a target efficiency constraint (latency or FLOPs). Similar to OFA, we use a 3-layer accuracy predictor common across all platforms. For latency, we use a lookup-table based latency estimator for Samsung Note 10 CPU provided by Cai et al. (2020). For other hardware platforms – namely NVIDIA GeForce RTX 2080 Ti GPU and the Intel(R) Xeon(R) Gold 6226 CPU – we measured actual latency with a batch size of 64 and 1 respectively. The estimated accuracies of the best models returned by this search were then verified on the actual ImageNet validation set.

Figure 3 reports the performances of accuracy-latency trade-off for CompOFA and OFA networks. We observe that the best model in CompOFA for evaluated latency targets are at least as accurate, despite their significantly smaller family size *and* training budget. This result validates our intuition behind the simple heuristic that aids in creating a smaller design space without losing out on Pareto-optimality or density of the generated models.

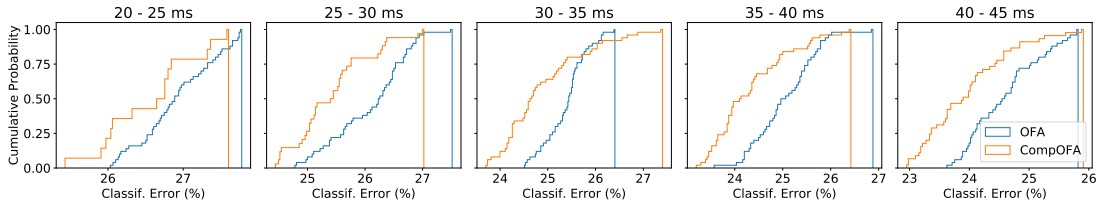

Figure 4: CDF comparisons of 50 randomly sampled models sampled in latency buckets of 5ms each. CompOFA has a higher fraction of its population at or better than a given classification error.

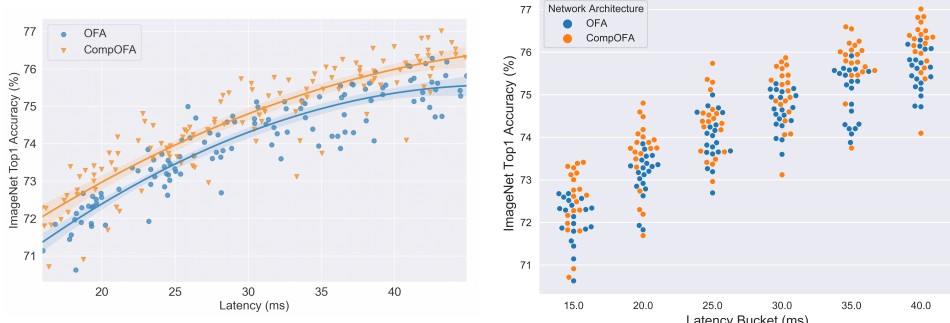

Figure 5: Random sampling of 20 models in each latency bucket of 5ms for CompOFA & OFA actual latency (left) and bucketed latency (right). CompOFA yields a higher average accuracy, i.e. as a population it has a higher concentration of accurate models. The shaded regions in the left plot show the 95% confidence interval of the average accuracy.

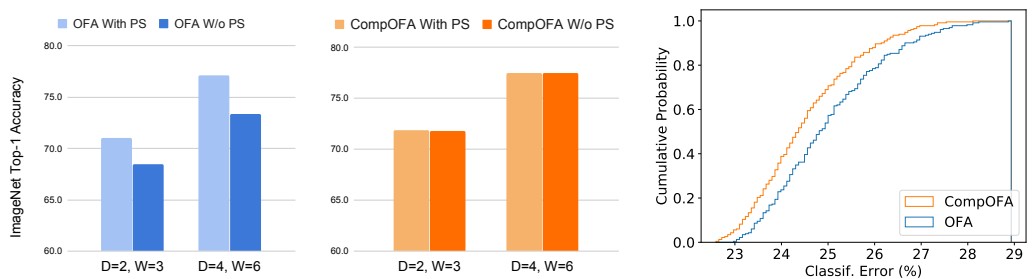

Figure 6: **Left and Center:** Comparing accuracies with and without progressive shrinking (PS). OFA suffers upto 3.7% accuracy drop when progressive shrinking is removed. For CompOFA, no such accuracy drop is seen when compared to the longer training with progressive shrinking.
**Right:** CDF of model configurations *common* to OFA & CompOFA. CompOFA does not lose accuracy despite its smaller cardinality and smaller training budget.

## 5.4 DESIGN SPACE COMPARISON

Apart from individual models, we further evaluate CompOFA at a population level through a statistical sampling of design space as introduced by Radosavovic et al. (2020; 2019). Using Samsung Note 10 as sample hardware, we divide the supported range of latencies into buckets of 5ms each. For each latency bucket, we randomly sample 50 models in OFA and CompOFA. Figure 4 plots the cumulative distribution function (CDF) of classification error for each of these latency buckets. In each bucket, the CDFs depict the fraction of models that exceed a given accuracy on the x-axis. CompOFA's CDFs lie above those of OFA, showing that CompOFA has a higher fraction of accurate models. Next, in Figure 5, we randomly sample 20 models in each bucket for each search space, and show that the *average* accuracy of a randomly picked model from CompOFA is higher than that in OFA. The takeaway of both these evaluations is that CompOFA yields a better *concentration* of accurate models and fewer sub-optimal models – i.e. a more accurate overall model population. Note that this is different from the *best* accuracy per latency target, where CompOFA matches OFA but with half the training budget.

## 5.5 EFFECT OF NUMBER OF PHASES

Cai et al. (2020) showed that OFA networks suffer a top-1 accuracy drop of up to 3.7% when progressive shrinking is removed, showing its role in resolving interference. With $O(10^{19})$ models interfering with each other for simultaneous optimization of their weights, a phased approach is needed to stabilize training by gradually train model configurations. In CompOFA, our primary method of reducing the training time is by reducing the number of phases in training. We repeat a

similar ablation to compare the effect of progressive shrinking in OFA & CompOFA. For both design spaces, we train with & without progressive shrinking and compare the accuracies of common sub-networks. Figure 6 shows CompOFA achieves the same accuracy with and without progressive shrinking. We attribute this to lower interference between sub-networks in a design space smaller by 17 orders of magnitude.

Next, we build a CDF of model configurations in CompOFA and then extract the same sub-networks from OFA. Figure 6 shows that CompOFA still maintains a slightly higher accuracy despite comparing the same models in both spaces. Thus, models discarded in the process of constraining the design space do not add to the accuracy of OFA or CompOFA – the retained models can achieve the same (if not better) accuracy independent of any potential collaboration from these discarded models.

### 5.6 Generalizing to other architectures

The guiding principle behind our heuristic – namely, the optimality of coupling depth & width over the unconstrained search space – is expected to apply to other architectures, datasets, and tasks. We demonstrate similar savings in training budget with our heuristic applied to a different base architecture.

We train OFA and CompOFA with the base architecture changed from MobileNetV3 to Proxyless-NAS (Cai et al., 2018). We keep the same depths and widths ($D = [2, 3, 4], W = [3, 4, 6]$) and build the search spaces of OFA & CompOFA using the cartesian product or the heuristic, respectively. In CompOFA, we fix the kernel size to 3 in all layers. Both networks use a width multiplier of 1.3.

The training hyperparameters, schedule, and search spaces are identical to those described in Section 5.1 and Table 1. Hence, CompOFA again requires half the training epochs – and thererfore half the training time, cost, and $CO_2$ emissions. Appendix A.2 shows CDFs of models common to both OFA-ProxylessNAS & CompOFA-ProxylessNAS, showing that the CompOFA has the same or marginally higher accuracies for the same model configurations despite half the training budget.

### 5.7 Search time

The intractable cardinality of OFA necessitated the use of latency estimators as a proxy to real latency measurement. With a significantly smaller design space, this need is removed – avoiding the time and effort of building these latency tables that eases "off-the-shelf" practical usability of CompOFA. We introduce a simple memoization technique during the evolutionary search to cache the measured latencies of model architectures and hence avoid its re-measurement, which is only practical with the smaller search space. Table 2 reports the average run time of NAS algorithms for a single latency target with this optimization added to both CompOFA and OFA, reducing the search time to just 75 seconds for CompOFA Fixed-Kernel. We also show that the evolutionary search converges in fewer iterations for CompOFA, in Appendix A.4

## 6 Conclusion

To conclude, we have built upon Once-For-All (OFA) networks to propose *CompOFA* – a design space for OFA networks using compound couplings between model dimensions, which speeds up the process of one-shot training and neural architecture search with hardware latency constraints.

We show that intractably large architectural search spaces are unnecessary for both accuracy and diversity of models. By leveraging the insight of compound couplings we introduced a simple heuristic that vastly shrinked the search space without losing on Pareto optimality. Despite its sparsity, this smaller search space is sufficiently dense to support the same diversity and range of deployment targets.

This smaller cardinality reduces interference in weight-shared training which allows CompOFA to reach the same accuracy in half the training budget. Once trained, CompOFA's tractability lends itself to easier extraction which is faster by 216x without the time or effort to build latency estimators. This improves the ease of "off-the-shelf" usability of our method in real-world settings. Finally, we apply our heuristic on another model and show that CompOFA continues to uncover similar gains.

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

## A  APPENDIX

### A.1  EXTRA MODELS IN OFA

Figure 8 depicts the stratification of the latency levels into multiple buckets of size 5ms each and uniformly sampled 100 sub-networks of OFA for each bucket. This box-plot helps in uncovering the considerable variance between the maximum and minimum accuracies of the models of the same latency bucket in OFA. This variance is mainly caused due to OFA's enormous search space that contains redundant models that do not improve the overall accuracy of the model distribution belonging to a certain latency bucket.

### A.2  COMPOFA-PROXYLESSNAS

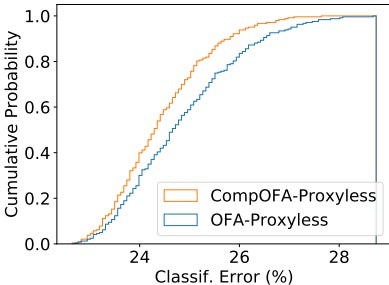

Figure 7: Cumulative distribution function of accuracies of model configurations common to OFA & CompOFA with the base architecture changed from MobileNetV3 to ProxylessNAS. Despite the change in architecture, the same heuristic allows CompOFA to train to the same or marginally higher accuracies with half the training budget.

## A.3 TRAINING SCHEDULE FOR COMPOFA-ELASTIC KERNEL

Table 3 shows the training schedule for CompOFA with Elastic Kernel. Compared to OFA, the training budget is reduced by 31%.

Table 3: CompOFA (Elastic Kernel)

| Phase | $K$ | $D$ | $W$ | $N_{sample}$ | Epochs | Wall Time | GPU Hours |
|---|---|---|---|---|---|---|---|
| Teacher | 7 | 4 | 6 | 1 | 180 | 28h 45m | 172h 30m |
| Elastic Kernel | 3, 5, 7 | 4 | 6 | 1 | 125 | 26h 51m | 161h 06m |
| Compound | 3, 5, 7 | 2, 3, 4 | 3, 4, 6 | 4 | 25 | 9h 21m | 56h 06m |
| Compound | 3, 5, 7 | 2, 3, 4 | 3, 4, 6 | 4 | 125 | 48h 01m | 288h 06m |
| **Total** | | | | | **455** | **112h 58m** | **677h 48m** |

## A.4 FASTER CONVERGENCE OF NAS

An evolutionary algorithm is set to converge after a certain number of iterations ($N$) beyond which the fitness value of the population ($P$) does not improve significantly. OFA runs NAS with a setting of $N = 500$ iterations for a population of size $|P| = 100$. Figure 9 demonstrates that the search time of NAS could further be reduced by reducing the number of iterations to $N = 300$ and $N = 50$ for CompOFA Elastic-Kernel and CompOFA Fixed-Kernel respectively, without losing out on their Pareto-optimality. Coupled with the removal of the latency predictor, these CompOFA-specific optimizations to the search are effective in reducing the search time and improving direct usability of CompOFA.

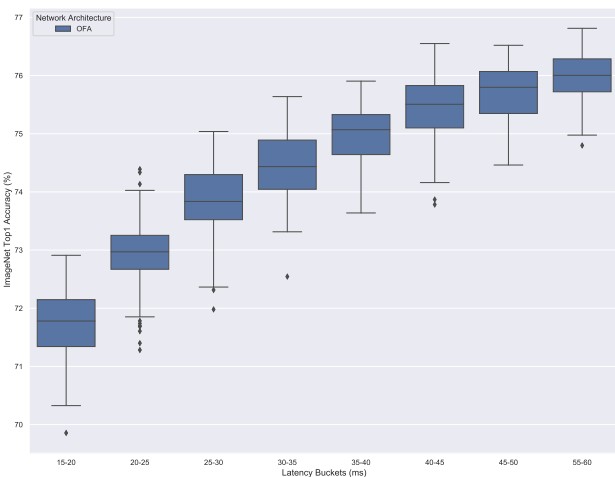

Figure 8: Distribution of accuracies of the randomly sampled models from OFA

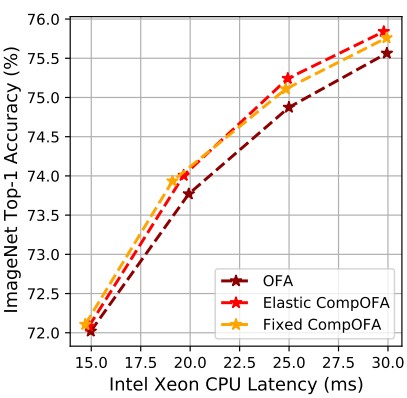

Figure 9: ImageNet accuracies and latencies on Samsung Note10 with reduced NAS iterations for CompOFA.

