# OpenReview forum: "CompOFA – Compound Once-For-All Networks for Faster Multi-Platform Deployment"
_ICLR.cc/2021/Conference — ICLR 2021 Poster_

### Official Review · AnonReviewer3 · 2020-10-28
**a heuristic for reducing architectural search time for compact models based on mobilenet architectures**

**Rating:** 6
**Confidence:** 3

**Review:**

Deploying models on multiple target end points with different hardware spec requires training model variants of an underlying architecture that meet the latency and other resource constraints on the device. OFA is a framework that helps this search for architectures like mobilenetv3 by training a model with large components and then training multiple instances of smaller models wherein components have reduced size so as to fit multiple target devices. Due to the large state space of parameter sizes for subcomponents, this search can take a while.

This paper proposes a heuristic to bring down the size. The main premise here is that searching for various subcomponents size parameters as if they were orthogonal design choices is not optimal. Good choices for parameters, for example the depth and the width are correlated. Therefore in place of searching as though these were orthogonal, the authors propose a correlated search to reduce the number of possibilities. This halves the overall search time, which is substantial saving given the time of the overall search.

The experiments are comprehensive on backing up the premise for the limited choice of data set and architecture considered. They show consistent improvements in overall training time  while showing the quality of resulting architectures is comparable to OFA methodology. This is done by showing the accuracy and size distribution of models obtained with this method vs OFA.

My concern with this paper is that the heuristic, while useful in this case, is not particularly interesting. Nor does the paper establish that this is applicable to other architectures. The result is that the scope of this paper is narrow. I would like more evidence to demonstrate that there are generalizable principles here.

---

> ### Author Response · Authors · 2020-11-17
> **Response to AnonReviewer3**
>
> Thank you for your constructive comment.
>
> 1. Our heuristic is grounded in insights from EfficientNet (Tan & Le 2019) & RegNet (Radosavovic et al 2020). EfficientNet showed that combining dimensions in simple ratios works well, and RegNet showed a quantized linear relationship is optimal. Both point to the insight that an increase along one dimension should accompany an increase along the other dimension. Our heuristic incorporates this simple relation.
>
> 2. This premise of our heuristic is still expected to hold for different architectures, datasets, and tasks as shown in these prior works. In our paper revision, we will add results our heuristic applied to a different base architecture, and show similar gains over the unconstrained search space.

---

### Official Review · AnonReviewer2 · 2020-10-28
**Pretty impressive results with a minimal (but important) search space modification**

**Rating:** 7
**Confidence:** 4

**Review:**

This paper presents a compounding strategy for constraining the search spaces for once-for-all (OFA) network training framework. This is motivated by observations in prior work (particularly the EfficientNet by Mingxing Tan and Quoc Le) that certain compound relations exist between network dimensions (widths, depths, resolutions) for achieving optimal accuracy-latency trade-offs. In this work, network depths and widths are compoundly coupled. The resulting method called CompOFA reduces the search spaces significantly from 10^19 candidates to 3^5 (CompOFA) or 10^10 (CompOFA-Elastic) candidates, thus cutting the training costs by 50% or 30%. The models found with CompOFA/Elastic for specific hardware latencies are also more accurate than those found with OFA, despite requiring less training costs.

Paper's strengths
- Both smaller search spaces and the avoidance of progressive shrinking further reduce the training costs of an already efficient framework (OFA) while not sacrificing prediction performance is impressive. This makes it more practically useful.
- The proposed compounding strategy is well-motivated by existing findings and nicely integrated to the OFA framework. It also clearly demonstrates that (human) prior knowledge is still very relevant in automated search for efficient networks.
- The models searched by CompOFA match the performance of (or in some cases, outperform) OFA models, both in terms of best models given certain latencies and sampled models at population levels.
- The paper is generally well-written and includes sufficient references to prior work.

Paper's weaknesses
- The pre-defined coupling configurations [D:2, W:3], [D:3, W:4], [D:4, W:6] are heuristically chosen based on the default search space originally proposed by the OFA paper. CompOFA works well because potentially these numbers happened to quite effective. In the more realistic settings (e.g., other datasets, tasks) where we do not have access to the prior numbers, it may be difficult to achieve good performance with a small number of coupling configurations, and OFA may work better to explore different combinations in a larger search space.

- The compound coupling only considers network depths and widths, and only a small modification to the design space is proposed to achieve that. If the proposed method considered image resolution, the paper would be more complete in the context of compound scaling and it would have a stronger contribution.

- In the OFA paper, smaller models are generated by shrinking the larger models, such that the smaller models (partially) retain the weights of the larger models. In this paper, all the smaller models are generated and trained at the same time. However, there is no information in this paper on how they are derived from the teacher network, e.g., whether they partially share the weights with teacher or they are distinctly initialized.

- While it is interesting that CompOFA models could outperform OFA models in some cases, this paper does not make it clear why this is the case. CompOFA's search space is supposedly a subset of OFA's search space. The models trained by CompOFA should be covered by OFA.

- Code is not provided in the submission for reproducibility and there is no promise of code release.

Minor comment
- Sec 5.2 mentions "by using just one stage of training after the teacher model" but the paper should explicitly mention that this applies only to CompOFA (fixed kernel). CompOFA (elastic kernel) in Table 4 actually still requires multiple stages.

This paper's ideas and contributions are nice-to-have but they are far from being groundbreaking.

##post-rebuttal##

I share the concerns of others reviewers that this paper has a limited novelty and narrowed scope but I think the authors have addressed other issues/concerns quite well. In my opinion, the key contributions mostly come from the insight and experimental results, and less so on the heuristic itself. Thus, I would adjust my rating to 7.

---

> ### Author Response · Authors · 2020-11-17
> **Response to AnonReviewer2**
>
> Thank you for your detailed & constructive review
>
> 1. We agree that these numbers may change across different base architectures but the premise of our heuristic -- i.e. the optimality of increasing depth & width together -- is still expected to hold for different architectures, datasets, and tasks as shown in EfficientNet and Radosavovic et al (2020). We will add another experiment in our revision to show similar gains with our heuristic applied to a different base architecture. We agree that algorithmically determining the search space would be ideal. This is a challenging independent problem and can be pursued outside the current scope of this paper. We believe our heuristics serve as a proof of concept and foundation for progress towards such methods.
>
> 2. We coupled depth & width as these are what influence search space cardinality, which in turn influences training interference and duration. On the other hand, resolution does not affect the number of unique architectures trained. So it allows one architecture to be used at multiple latency targets “for free” without increasing the training budget.
> We’ll make this distinction clearer in the paper revision.
>
> 3. There are 2 different accuracy evaluations -- best & average accuracy at a given latency target: \
>       a. __Average accuracy__ is _expected_ to be higher. We want to show that randomly sampling models in CompOFA yields better accuracies in expectation, signifying that our subspace is a better population with fewer sub-optimal models (Figures 3 & 4) \
>       b. __Best accuracy:__ Optimizing this (under given deployment constraints) is the end-goal of NAS. Hence we want to be able to match the best accuracy in all deployment scenarios as shown in Figure 2, despite the smaller training budget. Here, OFA & CompOFA are comparable throughout and the gap in their acc is <0.5%. \
> The reviewer is right that CompOFA’s search space is a subset of OFA’s. In Figure 5 we compare exactly this intersection and see that CompOFA’s CDF is comparable to same models in OFA, or marginally higher. While it could be argued that the marginal increase is potentially due to reduced training interference, our focus here was instead to establish that there’s no accuracy lost when discarding suboptimal models, allowing us to generate comparable models despite the smaller training budget.
>
> 4. Both CompOFA & OFA extract subnetworks of the teacher model in exactly the same manner, thereby retaining the shared weights between networks. We only differ in which subnetworks to extract and not how to extract them -- we sample a different configuration of depth/width. Once a configuration is picked, the mechanism of extracting a subnet is the same in OFA & CompOFA. We describe this in Section 3.1 and will make this clearer in the paper revision.
>
> 5. Thanks for the suggestion for code release. We will provide our anonymized code before the end of the rebuttal period.
>
> 6. We will add a clarification about CompOFA-Elastic Kernel in the revision, thank you for the suggestion. Note that the elastic-kernel variant is primarily shown only for completeness of comparison with OFA.

---

### Official Review · AnonReviewer1 · 2020-10-29
**This paper reduces the space of OFA with two tricks while still satisfying deployment requirements**

**Rating:** 5
**Confidence:** 2

**Review:**

#### Summary
This paper proposes to reduce the search space of OFA (a training scheme for obtaining networks for various deployment requirements) by scaling or shrinking the depth and width dimensions in NAS search space together. Because of the reduction, the multiple training phases of OFA is also less important and can be simplified to reduce training time.


#### Technical
* (+) The key observation made in this paper makes sense to me that the search space of OFA is over-sufficient and might be unnecessary for the target problem -- deployment in various environment with different latency/hardware constaints
* (+) the experiments of this paper looks good, the improvement over OFA in terms of training speedup and energy saving is impressive.

* (-)  I do appreciate the good engineering and intensive empirical results revealed by this paper but the technical novelty of this paper seems to me very increamental compared to OFa, though the arguments made in section 3.2 makes total sense me. For examples, the authors argued the model dimensions are not orthogonal, and the latency requirements only need to be satisfied in a way (granularity) that are below many thresholds. I think a better way to materialize these arguments is to explicitly characterize how these two observations should be incorporated into the architecture space, e.g., establishing direct correspondences between the design of the architecture space and the latency requirement, or by revealing how different dimension of architecture space are coupled (either theoretically or empirically ) and affect the design of the space. However, the paper ends up with a (not so intuitive) approach  -- that slightly modifies the space by coupling of two dimensions of the space, and starts discussing that by coupling these two dimensions, progressive multil-phase training can be eliminated to improve training.

---

> ### Author Response · Authors · 2020-11-17
> **Response to AnonReviewer1**
>
> Thanks for your constructive comment
> 1.  _“Establish direct correspondence bw arch space & latency target”_ \
> One small but important difference is that our search space should change with latency range, not a single target. We don't want to change the search space based on deployment target -- we want the same search space to support diverse targets (diverse hardware + latency target). This is the fundamental difference between OFA/CompOFA and traditional latency-objective NAS methods -- if changing the latency target necessitates rebuilding & retraining the search space then it defeats the purpose of OFA/CompOFA. \
> That said, we agree that it’s desirable to have systematic methods for determining the search space based on the range or granularity of latency requirement. Developing a method to translate latency requirements to search spaces is a separate problem on its own and can be pursued in future work, outside the current scope of this paper. We believe our heuristics serve as a proof of concept and foundation for progress towards such methods.
> 2. _“revealing how different dimension of architecture space are coupled”_ \
> Our simple heuristic is grounded in empirical insights from EfficientNet (Tan & Le 2019) & RegNet (Radosavovic et al 2020) which we repeat in the heatmaps. EfficientNet showed that combining dimensions in simple ratios works well, and RegNet showed a quantized linear relationship is optimal. Both point to the insight that an increase along one dimension should accompany an increase along the other dimension. We reaffirm this in our heatmaps in Figure 1. We show this coupling of model dimensions holds in OFA and incorporate into our heuristic. \
> We also believe this is the first work to demonstrate and apply these dimension couplings for families of architectures trained even with the added constraint of shared weights

---

### Official Review · AnonReviewer4 · 2020-10-30
**A good idea but poor motivation, execution, and presentation**

**Rating:** 4
**Confidence:** 4

**Review:**

Paper Overview:
This paper is aiming at optimizing the OFA method in neural network model searching and training. Though CNN models perform well for many prolems, there is a serious shortage of this method. Researchers need to build and train a new model for every new problem, which will cost a lot of time and resources. A recent work, OFA, proposed a new method to partly solve this problem by training a family of models at a same time by parameter sharing. However, the OFA still have some problems. The searching space of OFA is too huge so the training process still takes too long and cost lots of resources. Thus this paper addressed a new solution, CompOFA, to speed up the training and try to get a balance between the accuracy and latency by building constraints between dimensions of model searching space and "progressive shrinking" approach.

Strengths:
1. This paper gives out obvious evidence of their basic insight. They use a heatmap to show the trade-off relationship between the width and depth of different models.
2. This paper uses real data and SOTA works to do evaluation, makes their result more convincing.

Weaknesses:
1. The design idea and motivation of this paper is not well addressed. How do you decide to build the constraints? Why the trade-off is necessary? As far as I know, most of AI models care most about their accuracy, not latency, Unless the model is really too huge and slow (obviously the evaluation part of this paper only includes very small models with ms level latency). Because the inference speed can be easily increased by adding more GPUs. Since the inference job does not need to do synchronization, its is easily to achieve the linear scale out ratio. The paper just use a very arbitrary way to determine the constraints between these dimensions. How do you know that this is a good trade-off that researchers want?
2. The design part is too brief. The first subsection just tell us "We decide to build a constraint like this", but why? The second subsection about speeding up the training is too ambiguous. Using some pseudo-code or program chart can help readers to understand your work's logic and innovations.
3. The evaluation part is not solid enough to support this papers claim at the introduction. It claims to reduce searching space from 10^19 to 243. However, it still costs 50% training times compared to the original OFA method. And the training results also does not show apparent gains. The accuracy gain in most cases is lower than 0.4% while the overall accuracy does not exceed 80%. It will be better to add a classical light-weight CNN to be compared with, such as AlexNet, VGG or ResNet.

Other Comments:
This paper's idea is quite creative and valuable that adding some constraints to the searching space will help us to remove many unnecessary models and improve the overall efficiency. However, the current design of the solution is not mature enough. It is necessary to add theoretical proofs for important design choices and convince readers that the choice is reasonable. And the evaluation part could also be improved by adding some classic models to compare.

---

> ### Author Response · Authors · 2020-11-17
> **Addressing a number of technical/factual issues in this review**
>
> 1. __Motivation__ \
> A. _"most of AI models care most about their accuracy, not latency"_ \
> On the contrary, real-world applications care deeply about inference latency. Explosive growth in model size and computation cost of DNNs has necessitated research into efficient architecture design. It’s a well known fact that increasing model  computation complexity generally increases accuracy. This trade-off is well established and forms the very premise of entire bodies of work focused explicitly on latency/efficiency conscious DNNs which we highlight in our related work section. \
> B._"inference speed can be easily increased by adding more GPUs"_ \
> This statement is false, as the forward pass inference latency doesn’t change with the number of gpus. The reviewer is likely referring to increased throughput with more GPUs, which reduces mean latency over a large batch of data. However, the relevant definition of latency targeted by our work and related works is the inference time for a single forward pass of the network. This is also the latency of most relevance to model serving frameworks, such as TFServing and TorchServe. Finally, this statement ignores a vast majority of deployment scenarios where it’s not possible to add more GPUs/processors. \
> C. _How do you know that this is a good trade-off that researchers want?_ \
> Our work should be compared against OFA & other latency-aware NAS methods which have a well-defined objective: obtain trained models with best possible accuracies under given latency constraints. We do not propose a new architecture design that pushes the frontier of the accuracy-latency tradeoff. Instead, we focus on insights that enable refining the search space of OFA to only those models that lie on the Pareto-optimal accuracy & latency -- i.e. unequivocally better with higher accuracy for the same computational complexity.
>
> 2. __Design__ \
> A. _The first subsection just tell us "We decide to build a constraint like this", but why?_ \
> Our simple heuristic is grounded in insights from EfficientNet (Tan & Le 2019) & RegNet (Radosavovic et al 2020) which we repeat in the heatmaps (which this reviewer appreciated). EfficientNet showed that combining them in simple ratios works well, and RegNet showed a quantized linear relationship works well. Both point to the insight that an increase along one dimension should accompany an increase along the other dimension, and is confirmed in the heatmaps in Figure 1. Our heuristic incorporates this simple relation. \
> B. _speeding up the training is too ambiguous_ \
> Training details are in fact included in the sections we explicitly call out below If those are incomplete then we ask that the reviewer be more specific. \
> Section 4.1 decomposes the various factors affecting training budget, and argues that the source of training speedup will be reduction in number of phases, enabled by reduction in interference \
> Section 5.1 and Table 1 show the exact details & side-by-side comparison of what the search space looks like during this, how long it runs, and what LR schedule and other hyperparameters we use.

---

> ### Author Response · Authors · 2020-11-17
> **Addressing a number of technical/factual issues in this review  (contd.)**
>
> 3. __Evaluation__ \
> A. _it still costs 50% training times compared to the original OFA_ \
> We remind the reviewer that the original training time for OFA is 40-50 GPU days. At this magnitude, a 50% reduction is significant in relative & absolute terms. Our insight enables half the training budget -- a training time shorter by GPU weeks, training cost cheaper by ~$1k -- without sacrificing the quality of models for the end objective. Post training, our method also supports much easier extraction of models for a given target without needing to build latency predictors -- taking 5 minutes for CompOFA vs. 18 hours for OFA. These are significant reductions towards the practical usability of OFA. \
> B. _accuracy gain in most cases is lower than 0.4%_,  _better to add a classical light-weight CNN to be compared_ \
> We do not show accuracy comparisons with other works because we do not claim better accuracy -- we achieve the same accuracy but in a smaller training budget. Our contributions are the insights on search space design, and systems considerations that allow this reduction of training budget while maintaining accuracy and support for diverse deployment targets. Our experiments are focused on demonstrating these insights & speedups; not claiming accuracy gain. The best model generated by CompOFA matches that of OFA, and OFA already demonstrates that this accuracy is competitive against modern SOTA baselines such as MobileNets, EfficientNet, ResNets. \
> \
> It is expected that the best accuracy fetched by CompOFA (for a given deployment target) will match that of OFA since our search space is a subset. We show design space comparisons grounded in SOTA works (Radosavovic et al. 2019, 2020) to demonstrate that our search space has higher average (not best) accuracy of models -- implying a search space with fewer inaccurate models. We show ablations that demonstrate this subspace reduction is sufficient to bypass expensive progressive training, and this is indeed the source of our speedup. We confirm our systems motivations and show that we maintain the same latency support. We include comparisons that show how the post-training model extraction is simpler with our tractable search space. We will be happy to include more experiments if the reviewer can point to any specific issues with these claims.

---

### Author Response · Authors · 2020-11-25
**Paper revision, key contributions and clarifications**

We thank all the reviewers for their constructive comments. We restate our key technical contributions and insights:
- The number of degrees of freedom used to induce the model architecture search space can be reduced **without** loss of Pareto optimality
- Simple, easy to implement heuristics can be used to reduce the candidate pool size by several orders of magnitude
- **2x** training budget reduction compared to the state of the art to achieve a family of models along the same or better Pareto frontier with temporal granularity ~1ms.
- **216x** speedup on model search/extraction from the pre-trained super-model without latency estimators, as a direct benefit of the proposed search space reduction
- Demonstrated generalization to other base architectures.

We have uploaded a paper revision and supplementary material with the following key changes:

1. **Generalizability of heuristic:** A new experiment (Section 5.6) shows our heuristic applied to a different base architecture, namely ProxylessNAS. We apply the same heuristic to this architecture and show that CompOFA-ProxylessNAS is again able to match the accuracy of the unconstrained OFA-ProxylessNAS with only half the training budget of OFA, showing the generalizability of our idea to other architectures.
2. **Role of resolution:** We’ve added text in Section 4.1 clarifying the handling of resolution. Resolution does not affect the number of unique architectures, thus it does not hamper our training budget while still allowing one architecture to serve multiple latency targets. Thus, we deliberately keep an elastic resolution to add density to our design space **“for free”**.
3. **Code release:** We have made available our anonymized source code and pretrained models, with full instructions on training, evaluating, and searching for CompOFA networks at: https://github.com/compofa-blind-review/compofa-iclr21

Additionally, we highlight below some key clarifications from the individual replies to your reviews:

A. **Generalizability of heuristic:** \
The revision includes a new experiment showing the applicability of our heuristic to another base architecture. The core principle of CompOFA, i.e growing depth and width dimensions together to discard sub-optimal models, is based on prior established work and is expected to hold true across varying architectures, datasets, and tasks. We acknowledge that finding the optimal ranges of these dimensions based on deployment requirements would be ideal, but that is an independent and a more challenging problem for which our heuristics serve as a proof of concept. We aim to demonstrate that even with a simple heuristic proposed sizeable gains can be achieved.

B. **Accuracy comparisons:** \
CompOFA subspace identifies a better population of architectures by discarding lower-accuracy models. Therefore, CompOFA’s average accuracy (based on random sampling) is expected to be higher than OFA.
Since CompOFA’s architecture search space is a proper subset of OFA’s, the **best** accuracy is expected to be the same and we achieve our stated goal of matching the best model for ANY latency target, but with **half the training budget**. Some marginal (<0.5%) improvements in accuracy of common models in OFA could potentially be attributed to reduced training interference. We emphasize that accuracy improvement was NOT the stated goal of this work.

C. **Key clarifications/misunderstanding**
  - We emphasize that CompOFA does NOT aim to improve the state-of-the-art latency-accuracy Pareto frontier. It aims to match it at a fraction of the cost. [AnonReviewer4, AnonReviewer2]
  - We do NOT argue for trading accuracy for latency. CompOFA trains a family of models that populate the latency-accuracy tradeoff space, so that any point on the Pareto frontier of that space can be chosen based on application’s requirements for accuracy and/or latency. [AnonReviewer4]
  - We insist on the definition of latency as the amount of time taken by the forward pass of a candidate architecture. As such, this latency cannot be improved with more GPUs. We believe reviewer4 was referring to the mean latency (the inverse of throughput), which is the wrong definition of latency for this work as well as all works on latency-aware NAS. [AnonReviewer4]
  - CompOFA (+ OFA) aims at creating one architecture search space for all hardware and/or latency targets. [AnonReviewer1]

---

### Decision · Program_Chairs · 2021-01-07
**Final Decision**

**Decision:**

Accept (Poster)

**Comment:**

This paper focuses on  once-for-all (OFA) network training towards developing accurate models for different hardware platforms and varying latency constraints. The paper proposes an approach to significantly reducing model search space and thus training costs without losing in predictive performance. The paper is well-written, and the authors provide a thorough and convincing response to the concerns raised by some of the reviewers.